# Improvement of Plant Responses by Nanobiofertilizer: A Step towards Sustainable Agriculture

**DOI:** 10.3390/nano12060965

**Published:** 2022-03-14

**Authors:** Nosheen Akhtar, Noshin Ilyas, Tehseen Ahmad Meraj, Alireza Pour-Aboughadareh, R. Z. Sayyed, Zia-ur-Rehman Mashwani, Peter Poczai

**Affiliations:** 1Department of Botany, PMAS-Arid Agriculture University, Rawalpindi 46300, Pakistan; noshee.nawaz444@gmail.com (N.A.); zia.botany@gmail.com (Z.-u.-R.M.); 2College of Agronomy, Sichuan Agriculture University, Chengdu 611130, China; tehseenahmad55@hotmail.com; 3Seed and Plant Improvement Institute, Agricultural Research, Education and Extension Organization (AREEO), Karaj P.O. Box 3183964653, Iran; a.poraboghadareh@edu.ikiu.ac.ir; 4Institute of Genetics and Plant Experimental Biology, Uzbekistan Academy of Sciences, Tashkent Region, Tashkent 111208, Uzbekistan; sayyedrz@gmail.com; 5Finnish Museum of Natural History, University of Helsinki, FI-00014 Helsinki, Finland

**Keywords:** biofertilizer, nanobiofertilizer, nanoparticles

## Abstract

Drastic changes in the climate and ecosystem due to natural or anthropogenic activities have severely affected crop production globally. This concern has raised the need to develop environmentally friendly and cost-effective strategies, particularly for keeping pace with the demands of the growing population. The use of nanobiofertilizers in agriculture opens a new chapter in the sustainable production of crops. The application of nanoparticles improves the growth and stress tolerance in plants. Inoculation of biofertilizers is another strategy explored in agriculture. The combination of nanoparticles and biofertilizers produces nanobiofertilizers, which are cost-effective and more potent and eco-friendly than nanoparticles or biofertilizers alone. Nanobiofertilizers consist of biofertilizers encapsulated in nanoparticles. Biofertilizers are the preparations of plant-based carriers having beneficial microbial cells, while nanoparticles are microscopic (1–100 nm) particles that possess numerous advantages. Silicon, zinc, copper, iron, and silver are the commonly used nanoparticles for the formulation of nanobiofertilizer. The green synthesis of these nanoparticles enhances their performance and characteristics. The use of nanobiofertilizers is more effective than other traditional strategies. They also perform their role better than the common salts previously used in agriculture to enhance the production of crops. Nanobiofertilizer gives better and more long-lasting results as compared to traditional chemical fertilizers. It improves the structure and function of soil and the morphological, physiological, biochemical, and yield attributes of plants. The formation and application of nanobiofertilizer is a practical step toward smart fertilizer that enhances growth and augments the yield of crops. The literature on the formulation and application of nanobiofertilizer at the field level is scarce. This product requires attention, as it can reduce the use of chemical fertilizer and make the soil and crops healthy. This review highlights the formulation and application of nanobiofertilizer on various plant species and explains how nanobiofertilizer improves the growth and development of plants. It covers the role and status of nanobiofertilizer in agriculture. The limitations of and future strategies for formulating effective nanobiofertilizer are mentioned.

## 1. Introduction

Around 800 million people suffer from chronic hunger worldwide, and this figure may rise with time. Moreover, humans suffer from many health issues due to unstable and low-quality food [1]. The nutritional quality of the crop is linked with climate change, as unpredictable seasonal and spatial variations disturb crops’ life cycles. Various biotic and abiotic stresses affect the crop’s production and nutritional quality. Ultraviolet radiation damages plant cells and disturbs the structure of cell organelles. Drought stress directly decreases the rate of photosynthesis in plants. It also poses negative effects on cell enlargement, cell division, and differentiation [2,3]. The weathering of parent rock material, industrial waste and sewage sludge discharge, and uncontrolled use of heavy-metal-containing agrochemicals cause high soil salinity, heavy metal contamination, etc. [4,5]. These agrochemical fertilizers end up in water bodies and cause eutrophication. The long-term use of agrochemicals has negatively impacted soil health and human health. This has been a major cause of soil and water pollution, soil erosion, nutrient imbalance, lessened agrobiodiversity, decreased soil fertility, and low water holding capacity, and has caused disturbances in natural soil flora and fauna [6]. The use of excess nutrients is polluting soil, and water scarcity makes the condition worse [7]. It was reported that a large portion of synthetic nutrients are not absorbed by plants, and instead run off from the fields and become integrated into water resources. The excess amount of synthetic nutrients causes serious concerns about food safety as well as its quality. There is a need to focus on the development of sustainable agricultural practices. We cannot replace the use of chemical fertilizers, but we can use other strategies so that they have beneficial effects. Plants are facing various stresses in their natural habitat that reduce their growth [8,9]. Plants undergo different morphological, physiological, and biochemical changes during stress conditions. Plants face several stresses such as drought, salinity, freezing stress, heat stress, and heavy metal stress. When plants are exposed to adverse conditions, the rate of formation of reactive oxygen species (ROS) becomes high, which causes oxidative stress, reduces the membranes’ stability, and affects the structure and functions of organelles in the plant cells. If the stress occurs at the anthesis stage, the ovaries are aborted, and the fertilization rate is reduced [10]. There is a need to develop environmentally friendly strategies for the amelioration of stresses and to increase the yield of crops. This can be achieved by enhancing the productivity of crops and by expanding the area under cultivation. The former approach is more suitable, and can be attained through biofertilizers. Biofertilizers release various metabolites, plant hormones, and polysaccharides that improve soil quality and plant growth. They have inhibitory effects on the growth and proliferation of different plant pathogens. They create tolerance or resistance in plants against biotic stress [11,12,13]. The use of plant-growth-promoting rhizobacteria (PGPR) as biofertilizers has been suggested as one of the most suitable and sustainable approaches for improving crop yield. They enhance the growth of plants by a variety of direct and indirect mechanisms. Direct mechanisms include nitrogen fixation, production of siderophore, phytohormone, and exopolysaccharide, and solubilization of phosphorus and potassium. Their inoculation in the rhizosphere makes the plants stress-tolerant or stress-resistant. Their use also minimizes the dependency on chemical fertilizers [14]. Application of *Azospirillum* sp. improves the germination and morphological characteristics of *Triticum aestivum* L. under abiotic stresses [15]. Inoculating PGPR decreases electrolyte leakage and increases the relative water content and proline synthesis in wheat [16] and soybeans [17]. Pod fresh weight, pod number, and seed weight were also significantly improved by biofertilizer compared to untreated plants under drought stress [18].

Nanotechnology is a new approach that is gaining attention in the agricultural sector of developing countries. Nanoparticles act as stimuli that activate various defense systems in plants facing unfavorable conditions. The use of nanoparticles is superior to using common salts as a source of fertilization due to their high surface area, high solubility, and lightness. Nanobiofertilizer is produced through the combination of nanoparticles and biofertilizers. It is a technique in which biofertilizers are encapsulated within a suitable nanomaterial. They release the nutrients into the soil in a controlled manner and reduce the side effects of environmental stresses. They decrease the use of chemical fertilizer, improve the availability and uptake of the nutrients, are eco-friendly, and are cost-effective (Figure 1) [19,20]. It was observed that nanoparticles have direct and indirect effects on plant–microbe interaction; direct effects include the availability of nutrients in the rhizosphere, while indirect effects include the stimulating effects on bacterial strains [21]. It was reported that the use of only 75 ppm CeO_2_-nanocomposite significantly enhanced the length of *Trigonella foenum-graecum*; the use of a very small amount gave pronounced results [22]. Nanobiofertilizer has the dual properties of bioinoculants and nanoparticles that enhance the chances of crop establishment. They maintain the delivery of nutrients at the target destination. Due to a lack of information about the interactive role of biofertilizers, nanoparticles, and plants, nanobiofertilizers still have not been widely promoted [23].

There is a need to work on and explore more suitable nanobiofertilizers so that they will be commercialized and available to farmers. Nanobiofertilizer is still not commercialized on a large scale. This article highlights the importance and primary mechanism of actions of nanobiofertilizer on plants. The main objective of this review is to evaluate the role of nanobiofertilizer in plants facing biotic and abiotic stresses.

## 2. Constituents of Nanobiofertilizer

Nanobiofertilizer constitutes two main components: (1) nanoparticles and (2) biofertilizer.

### 2.1. Nanoparticles

Nanoparticles are microscopic particles, usually ranging from 1–100 nm. The commonly used nanoparticles for the formulation of nanobiofertilizer are silicon, zinc, copper, iron, and silver. The uptake and accumulation of nanoparticles in plants depends on their chemical composition, shape, size, and agglomeration state. The translocation of nanoparticles is inversely proportional to their size. It is also linked to the plant species, as the receptors vary from plant to plant. A plant may act as an accumulator for one type of nanoparticle and an excluder for another kind of nanoparticle. Different nanoparticles use different mechanisms to improve the growth of plants [24,25]. Figure 2 represents the application of nanoparticles and their movement inside plants. When the nanoparticles are applied in the rhizosphere they enter in plant roots by endocytosis, through carrier proteins, or by plasmodesmata. When they are applied in the form of foliar spray they simply diffuse through the stomata and enter the vascular bundles. They also travel inside the plant by following symplastic and apoplastic pathways. The pathway of the nanoparticles’ movement depends on the type of plant and type of nanoparticles [26,27]. The detail of nanoparticles used in the synthesis of nanobiofertilizer is as given below.

#### 2.1.1. Silicon Nanoparticles (SiNPs)

Silicon is present in the soil in ample amounts and is categorized as a beneficial substance for plants. It is present in the form of mono silicic acid, having the formula Si (OH)_4_. The extensive use of agricultural lands decreases the amount of silicon in soil and causes its deficiency in plants [28]. According to an estimate, 210–224 million tons of silicon are removed from the fields each year. The application of silicon enhances metabolite production and improves the growth of plants. It develops resistance or tolerance in plants under stress conditions. There are Si transporters in plants that transport Si against the concentration gradient. It was reported that SIT1 allows the movement of Si to cortical cells, while SIT2 transports Si from cortical cells to xylem. The uptake and translocation of silicon in plants depends on various factors. It also depends on the type of plant species. Some plants are silicon accumulators while other are nonaccumulators. In silicon accumulators, the uptake of Si is ˃1% of the plant’s dry biomass, whereas nonaccumulators have a Si content <1% [29,30].

Silicon is considered as quasi-essential element. It controls gene expression in a positive way that makes the plant stress-tolerant [31]. It reduces the negative effects of oxidative stress in plants and enhances the uptake and translocation of nutrients from roots to aerial parts of plants. It has been documented that the plants that receive the application of silicon show a reduction in the rate of transpiration and enhancement of the rate of photosynthesis. It also reduces the damaging effects of insects and pathogens [32]. It was observed in a study that the application of silicon reduced the damage caused by the leaf folder and stem borer in *Oryza sativa* L [33]. It has also shown positive results on plants facing drought stress. The plants that received the exogenous application of silicon showed a significant increase in the production of antioxidants and growth hormones [34].

Silicon nanoparticles perform better as compared to Si salt. Green synthesis of nanoparticles is better, and it can be performed by using plants or their metabolites. Bioactive agents such as microbes and biowastes can be used. Green synthesis of SiNPs was performed by using *Bambusa bambos* (L.) Voss. The stem was cut into pieces and exposed to pyrolysis. After pyrolysis, it was treated with hydrochloric acid, sulphuric acid, and nitric acid, and then calcinated in an electric furnace [35]. *Azadirachta indica* A. Juss. leaves were also used to synthesize SiNPs. The leaves’ extract can reduce the size of Si. Extraction of SiNPs was performed from rice husk. Various characterization techniques were used to evaluate the characteristics of nanoparticles [36,37].

The literature has documented that the exogenous application of SiNPs gives better results than Si salt. They increase the activities of ascorbate peroxidase, glutathione reductase, dehydroascorbate reductase, superoxide dismutase, and ascorbic acid in *Zea mays* L. facing heavy metal stress [38]. Silicon nanoparticles also alleviate UV stress more efficiently than Si salt in *T. aestivum*. Silicon improves the activities of antioxidant enzymes such as viz. guaiacol peroxidase, ascorbate peroxidase, catalase, and superoxide dismutase. It also enhances the content of nonenzyme antioxidants such as proline, phenolics, flavonoids, and ascorbate. Improved synthesis of antioxidants helps scavenge reactive oxygen species and maintain the rate of photosynthesis in plants [39]. Foliar spray of Si and selenium nanoparticles on *Fragaria vesca* L. improved relative water content under drought stress. These treatments also enhanced the anthocyanin, ascorbic acid, and total phenolic compounds in plants [40]. SiNPs decrease the uptake and accumulation of heavy metals in food crops. They immobilize the metals in the soil and compete with the heavy metals for absorption and translocation in the plant. They also enhance the growth by translocating nutrients such as manganese, sulfur, phosphorus, and potassium from the soil to aerial parts of plants [33]. Additionally, they increase fresh weight, dry weight, chlorophyll a, and chlorophyll b content of *Ocimum basilicum* L. by osmotic adjustment under salinity stress [41,42].

#### 2.1.2. Zinc Nanoparticles (ZnNPs)

Zinc (Zn) is an integral part of various enzymes in plants. It is a micronutrient that activates different enzymes involved in carbohydrate metabolism. It plays a significant role during the transcription of DNA and acts as a co-factor of enzymes. It plays a vital role in biochemical pathways concerned with photosynthesis, auxin, protein, carbohydrates, and starch metabolism. It also maintains the structure and functions of biological membranes. Additionally, it adjusts the potassium level in the cells and regulates the opening and closing of stomata [35].

Zn deficiency inhibits tillering and causes chlorosis and stunted growth of crops. It inhibits pollen production, affects pollen tube growth, and reduces the rate of fertilization and seed set. It causes structural and biochemical changes in the extracellular matrix of pollen grains and downregulates esterase [36,37]. The application of Zn decreases the photo-oxidative damage and activates antioxidant enzymes that scavenge reactive oxygen species. Application of different levels of Zn fertilizer on *T. aestivum* showed that 11.4 kg/ha Zn concentration significantly improves root surface area, shoot dry weight, dry root weight, and root length [43,44]. In another field experiment, application of Zn at the rate of 15 kg/ha accelerated plant height, tasseling, and silking of *Z. mays*. It also improved grain yield and protein content [39]. The use of Zn enhances the growth of plants under unfavorable environmental conditions [45]. Foliar application of Zn fertilizer improves the quality and quantity of crops, but excessive use of chemical fertilizers is causing an imbalance and decreases the growth and yield of crops [46].

Zn nanofertilizer slowly releases the nutrients in a controlled way and has high translocation in plants. When the salts are converted to NPs, their physical and chemical characteristics are changed, and their application on plants produces better results [47]. Green synthesis of ZnNPs was done using the leaf extract of *Camellia sinensis* (L.) O. Kuntze. Zinc acetate was dissolved in the fresh plant extract and placed on a magnetic stirrer followed by centrifugation. The ZnNPs were washed and dried at 80 °C [48]. ZnNPs having the size of 18 nm were prepared by treating zinc acetate with sodium hydroxide and thioglycerol [49].

The application of ZnNPs reduces the adverse effects of biotic and abiotic stresses. It detoxifies heavy metals, increases water use efficiency, maintains membrane stability, and balances the uptake of nutrients. Different levels of ZnNP application increase the yield of *Sorghum bicolor* (L.) Moench. by 22–183% under drought stress compared to untreated plants [50]. They also enhance the DPPH (2, 2-diphenyl-1-picryl-hydrazyl-hydrate) radical scavenging activity in the root and shoot of *Camelina sativa* L. grown in saline soil [51]. When Zn and Fe nanofertilizers were applied in the form of foliar spray, they improved the morphological attributes and increased the yield and yield components significantly. They enhanced the yield of *Cicer arietinum* L. to 34% compared to the control [52]. It was also observed that the exogenous application of ZnNPs improves the anatomical structures of the leaf and stem and enhances the rate of photosynthesis. The yield of *Solanum melongena* L. was also increased from 50% to 66% grown in saline soil in the presence of a limited water supply [53]. The number of fruits, their weight, and size significantly increased in *Mangifera indica* L. when treated with Zn and SiNPs under salinity stress. This can be linked with the role of Zn in the development of floral parts, and the synthesis and balance of phytohormone content in plants also plays an important role [48].

#### 2.1.3. Copper Nanoparticles (CuNPs)

Copper (Cu) is an essential nutrient required for normal plant growth. It has a structural and functional role in plants. It is a micronutrient, and its content in the plant depends on its mobility and bioavailability in the soil [49,50]. Plants maintain the amount of Cu at the optimum level in various tissues. It is involved in various processes inside the mitochondria and chloroplast. Its functions include protein trafficking, mitochondrial respiration, iron mobilization, hormone signaling, and cell metabolism [54,55]. Cu is a structural component of plastocyanin. It enhances the production of hydrogen peroxide, which is a signaling molecule and develops stress tolerance or resistance in plants. It decreases plant wilting by providing strength to the cell wall. The exogenous application of Cu enhances biomass production in plants and improves plant yield and quality as well [56,57].

The deficiency of Cu in plants results in leaf curling, stunted growth, and chlorosis, and disturbs the process of photosynthesis. Cu deficiency enhances the production of reactive oxygen species. It affects photosystems and the flow of electrons in the electron transport chain. Cu deficiency also decreases nitrogen and carbohydrate metabolism [58].

The deficiency of Cu can be overcome by the exogenous application of Cu on plants. The application of CuNPs improves the growth and yield of crops. CuNPs can be synthesized by using leaf extract of *A. indica*. The leaf extract was added dropwise in a cupric chloride solution under continuous stirring and observed for color change from pale yellow to dark brown. The color change indicates the formation of CuNPs. The solution was centrifuged and CuNPs were dried after washing. Different characterization techniques such as Fourier Transform Infra-Red (FTIR), Scanning Electron Microscope (SEM), Transmission Electron Microscope (TEM), Zetasizers, and X-ray Diffraction (XRD) were used to confirm the synthesis and purity level of CuNPs [59]. The fruit extract of *Citrus limon* (L.) Burm. f. was also used to reduce the size of copper salt to nano-scale [60]. It was also reported in the literature that the leaf extract of *Tinospora cordifolia* (Willd.) Miers. [61], *Eclipta prostrate* (L.) L. [62], and *Plantago asiatica* L. [63] were used to synthesize CuNPs.

The rapid absorption and slow release of CuNPs make them an excellent candidate to be used. It was documented that the application of Cu and ZnNPs improved the morphological and physiological parameters in *Ocimum basilicum* L. [64]. It improves water status, maintains the rate of photosynthesis, and enhances the activities of antioxidants to create drought tolerance in crops. Foliar application of CuNPs improves the fruit quality of *S. lycopersicum*. It also enhances the synthesis of bioactive compounds and catalase and superoxide dismutase activity [65].

#### 2.1.4. Iron Nanoparticles (FeNPs)

Iron is one of the essential micronutrients required for the growth of plants. It is a co-factor and acts as a catalyst in various biochemical reactions. It is required to synthesize different enzymes and regulate photosynthesis and respiration. Iron deficiency reduces the synthesis of chlorophyll, and the leaves turn yellow. It causes necrosis and chlorosis in the leaves. The application of iron as a fertilizer improves plant growth and reduces the damaging effects of environmental stresses [66,67].

The application of nanofertilizer reduces the adverse effects of chemical fertilizer, and their application using both the soil and foliar method is helpful for improving the growth of plants [68]. Green synthesis of FeNPs is carried out using plant extract. The commonly used plants are *C. sinensis*, *A. indica*, and *Eucalyptus tereticornis* (Sm.) [69]. The phytochemicals of plant extract reduce the size of Fe salt and stabilize the reaction. The commonly used FeNPs are iron oxide hydroxide, ferric oxide, ferrous ferric oxide, and iron mineral complex [68]. The use of FeNPs generates positive responses when applied to plants under stress [70]. Plants absorb them quickly, and they reduce the toxic effects of ions. Fertilization using FeNPs increases the content of phenol, vitamin C, and glutathione in *S. lycopersicum* [71]. The application of FeNPs induces selective uptake of substances from the root membrane. It reduces the absorption of sodium, which enhances the potassium content in shoots and makes the plant salinity-resistant. The combined application of FeNPs and salicylic acid induces drought tolerance characteristics in *F. vesca*, as they show good results when applied to the in vitro culture of plants [72]. FeNPs stimulate the germination of *T. aestivum* and increase the root length and shoot length compared to untreated plants [73]. It has been reported that Cd stress reduces the rate of photosynthesis, causes oxidative stress, and reduces the yield of *T. aestivum*, having a high concentration of Cd in the grains. The application of FeNPs minimizes the uptake of Cd and enhances the Fe content in the grains and tissues of the plant. It also improves growth by suppressing the adverse effects of Cd stress on plants [74].

#### 2.1.5. Silver Nanoparticles (AgNPs)

Silver ions play a significant role in shoot formation, root formation, somatic embryogenesis, and genetic transformation. The application of silver decreases the synthesis of ethylene in plants and reduces the chances of chlorophyll destruction and chlorosis. It also enhances the production of secondary metabolites, downregulates the process of aging, and improves growth and grain yield in crops [73,74].

The role of nanoparticles in plants depends on their properties, such as chemical composition, surface reactivity, and application rate [75,76]. A high concentration of nanoparticles causes damaging effects. At the same time, the appropriate dose develops a positive response in plants, and the amount of dose is linked with the type of plant species. Nanoparticles are more effective, as documented in the literature [77]. Several plants have been used for the green synthesis of AgNPs due to their active Phytoconstituents, which reduce the size of Ag salt. These metabolites include aldehydes, flavonoids, ketones, tannins, carboxylic acid, amides, and terpenoids [78]. The leaves of *Ocimum tanuiflorum* L., *Syzygium cumini* (L.) Skeels., *Centella asiatica* (L.) Urban., and *Citrus sinensis* (L.) Osbeck. (peel) were boiled, and the resulting extract was slowly added to the solution of silver nitrate (1 mM). The change in color from yellow to dark brown indicated the synthesis of nanoparticles. The characterization was performed using UV-Visible spectroscopy, Atomic Force Microscopy, Scanning Electron Microscopy, and X-rays Diffraction [79].

The application of AgNPs (synthesized using *Eucalyptus globules* Labill.) improved the germination and antioxidant activity in the *Allium cepa* L., *Z. mays* and *Trigonella foenum-graecum* L. [80]. The use of AgNPs in the culture of *Dianthus chinensis* L. reduced the hyper-hydricity and hydrogen peroxide production to 13% and 50%, respectively [81]. They also reduced the production of reactive oxygen species and improved the length of roots in *G.max* [82]. They increased the carotenoid, chlorophyll, and polyphenol oxidase content to 46%, 45%, and 33%, respectively, in *S. lycopersicum* infected with *Alternaria alternate*. The decrease in disease severity was 58% compared to untreated plants [83]. Water uptake is not consistent in plants while leaves actively absorb the nutrients applied in the form of foliar spray. It is an inexpensive method, and more feasible as compared to soil application [84]. The foliar application of AgNPs increased the fresh weight and chlorophyll content to 32% and 23%, respectively, in *S. lycopersicum* infected with *A. solani*. The decrease in fungal spore count was 48%, and the activity of stress enzymes, which create stress tolerance in plants, was enhanced [76]. They also improved the rate of seed germination, root length, shoot length, and biomass in *Satureja hortensis* L. facing salinity stress [85,86]. The modulation of different genes expression in plants in response to the application of nanoparticles is presented in the table below (Table 1).

### 2.2. Biofertilizer

Biofertilizer has beneficial microbes such as blue-green algae (BGA), plant-growth-promoting rhizobacteria (PGPR), fungal mycorrhizae, or processed plant material [97]. The concept of biofertilizer can be traced back to ancient Roman writings and classical Greeks describing different agricultural practices to improve the yield of plants. The use and marketing of microbes as biofertilizers began in the late nineteenth century [98]. The extensive use of chemical fertilizer is presently polluting the agroecosystem. It is one of the major sources of soil pollution, water pollution, and air pollution. Developing a cheap, nontoxic, and eco-friendly alternative to obtain high productivity in agriculture without any side effects is the need of the hour [99]. The biofertilizers are eco-friendly, cost-effective, sustainable, and have long-lasting effects. They colonize the rhizosphere of plants and improve their growth by enhancing nutrient availability [100]. Figure 3 represents the different direct and indirect response of rhizobacteria to make the plant stress-tolerant or -resistant.

They release plant-growth-promoting substances, perform phosphate [101,102] and potassium solubilization [103], and fix the atmospheric nitrogen in the soil [104]. Some biofertilizers release a siderophore that sequesters Fe [105,106], making it unavailable for pathogenic organisms and inhibiting the growth of pathogens. They produce various phytohormones such as indole-3-acetic acid, gibberellins, cytokines, abscisic acid, etc. [107], which help in plant growth and development [108,109]. However, the issues with biofertilizers include their high dose requirement to cover a large area, low quality, low performance in a fluctuating environment, and substandard shelf life [110,111]. The use of nanobiofertilizer has overcome many issues, and agriculture relies on nanotechnology to improve the growth and yield of crops and maintain the natural ecosystem. They enhance the slow release of fertilizer and maintain its quality. Biofertilizers are a renewable approach that reduces the cost of chemical fertilizer and has long-lasting effects on the field [11].

## 3. Formulation of Nanobiofertilizer

Nanobiofertilizer is a combination of nanoparticles and biofertilizers. It can enhance the nutrient use efficiency of plants by slowly releasing nutrients. It can improve soil properties by imparting long-lasting effects on the soil’s physical, chemical, and biological properties. Formulation of nanobiofertilizer includes encapsulation of nanoparticles with biofertilizer. It protects the bacterial strains from mechanical stress, and the slow release of nutrients further enhances the efficiency of this product. Encapsulation is a process of accommodating biofertilizer cells in the nanomaterial capsule. It involves the use of a non-toxic, biodegradable material, such as calcium alginate, and starch. Starch increases the growth of bacterial strains [11,97].

The synthesis of nanobiofertilizers involves three essential stages: (1) the growth of biofertilizer culture; (2) its encapsulation with the nanoparticles; and (3) the evaluation of its effectiveness, quality, purity, and shelf life [112]. Nanobiofertilizer can also be prepared in the form of a microcapsule. Its synthesis involves the mixing of suspension of PGPR with the mixture of 1.5% sodium alginate, 3% starch, and 4% bentonite in the ratio of 2:1. The mixture is coated with the crosslinking calcium chloride solution, and the microcapsules are washed with sterile distilled water [113]. A combination of nanoparticles and salicylic acid has also been used to prepare nanobiofertilizer. In this method, the biofertilizer is mixed with sodium alginate (2%), ZnONPs (1 µg/mL), and salicylic acid (1.5 mM). The solution is coated with calcium chloride (3% solution) and 1-mm beads are formed, air-dried, and incubated at 4 °C [114]. It was reported that organic waste such as flowers, cow dung, and kitchen waste can also be combined with the nanoparticles to make an effective nanobiofertilizer that enhances soil fertility. The organic waste was washed with water to remove impurities, chopped into small pieces, and subjected to decomposition or pyrolysis. This partially decomposed or pyrolyzed waste was combined with nanoparticles to make nanobiofertilizer [115].

## 4. Nanobiofertilizer for Improving Soil Fertility

Today, the uncontrolled use of chemical fertilizers disturbs soil structure, pollutes water resources, causes eutrophication, soil toxicity, and nutrient leaching, and has a toxic effect on the ecosystem and human health. There is a need to work on the management of soil health for productive and sustainable agriculture. Soil nutrients should be managed properly to meet the demands of crops without any adverse effects on the ecosystem [116,117].

Nanobiofertilizer improves soil fertility by various strategies including: (a) restoring nutrients by nitrogen fixation and phosphorus solubilization; (b) enhancing nutrient absorption capacity of the soil; (c) siderophore production that chelates iron and sequesters heavy metals to make them unavailable for plants; (d) producing phytohormones and compatible solutes; (e) conserving soil moisture; and (f) developing resistance or tolerance in plants against biotic and abiotic stresses. They also enhance the nutrient use efficiency of plants [118,119].

## 5. Plants’ Responses to Nanobiofertilizer

The use of nanobiofertilizer improves plant growth significantly. The coating of nanoparticles with biofertilizers enhances the efficiency of biofertilizers and causes the sustained and slow release of nanoparticles in the plant’s rhizosphere. They maintain the dissolution of fertilizer and reduce the chances of leaching [120]. They also enhance the quality of crops, i.e., the production of secondary metabolites such as enzymatic (catalase, superoxide dismutase, and peroxidase) and nonenzymatic antioxidants (Phenols and flavonoids) is increased. These metabolites increase the shelf life of vegetables and fruits and have various health benefits [121]. The synergistic mechanism of action of biofertilizer and nanoparticles gives intensified responses when applied to plants. It activates different mechanisms in plants responsible for the better development and yield of plants. They also mitigate the adverse effects of toxic chemicals and reduce or inhibit the growth of pathogens in the rhizosphere of plants (Table 2). Nanobiofertilizers aid in bioremediation and replenish essential nutrients in the soil. They upregulate genes responsible for producing antioxidants, osmolytes, and stress-related proteins, reduce the damaging effects of ROS on plants, and maintain the structure and function of the cell. They also preserve membrane transporters’ enhanced hormonal production and their activities [122,123]. Figure 4 shows the response of plant cells under water stress and the effect of the application of nanoparticles and biofertilizers. When the plants receive these treatments, their antioxidant systems become active, protecting the cell membrane and organelles from the adverse effects of stress. They also increase their levels of growth hormones, including indole acetic acid and cytokinin, while reducing the production of stress hormones (abscisic acid). These changes make the plant stress-tolerant and increase the chances of crop establishment under unfavorable environmental conditions [124].

The application of nanobiofertilizer increases the number of grains per comb, the number of rows per comb, grain yield, and biomass yield and improves the harvest index of *Z. mays* in a limited water supply [136]. The use of gold nanoparticles accelerates the performance of *Bacillus subtilis* (33%), *Pseudomonas fluorescens* (57%), and *P. elgii*. They also improve the plant’s growth when used in the formulation of nanobiofertilizer [137]. It has been observed that biofertilizer (PGPR) controls the availability of nutrients in the soil, including nanoparticles. An experiment was designed to evaluate the role of nanoparticles on *S. lycopersicum* with and without *Azotobacter salinestris* inoculation. *Azotobacter salinestris* synthesized exopolysaccharides (EPS). This EPS entrapped the nanoparticles, maintained their concentration in the rhizosphere, and minimized the adverse effects of the high level of some nanoparticles [138]. Inoculating *Bacillus fortis* + ZnNP in *Cucumis melo* L. under Cd stress has enhanced photosynthetic pigments, plant biomass, flavonoid content, phenolics, transpiration, and stomatal conductance. It reduces the uptake, translocation, and accumulation of Cd metal in plants [139]. The combined application of SiNPs and PGPR on *Z. mays* grown under drought stress improved the rate of photosynthesis, relative water content, and activity of antioxidant enzymes. They also caused an increment in the yield attributes, increased nutrient uptake, and translocation in plants [140]. Application of FeNPs and *Bacillus subtilis* in *Cucurbita moschata* Duchesne decreased the adverse effects of arsenic toxicity. It improved photosynthesis and gaseous exchange and increased the production of superoxide dismutase and catalase [141].

The co-application of ZnO NPs and biofertilizer decreases the level of Na^+^ by 60% (roots) and 74% (leaves) in *Carthamus tinctorius* facing salinity. They also increase the number of pods per plant, number of leaves, and capitulum weight. These changes can be linked with the role of Zn, as it is required for chlorophyll production and auxin biosynthesis and plays an essential role in germination, soil fertility, and pollen performance. The combined application of ZnO NPs and biofertilizer in plants produces significant results, which shows their collaboration [142]. The interaction of biofertilizer and nitrogen NPs increases the available potassium, phosphorus, and nitrogen levels in the soil by 126%, 283%, and 38%, respectively. They interact with the soil and roots of plants and enhance the availability and uptake of nutrients. Biofertilizers, including rhizobacteria, release nutrient-dissolving organic acids such as citric acid, propionic acid, lactic acid, ketogluconic acid, succinic acid, and gluconic acid. They also fix atmospheric nitrogen and make the soil fertile. They increase the rate of photosynthesis, and the analysis of the plant’s part showed a high level of nutrients, including potassium, phosphorus, and nitrogen [143].

## 6. Current Scenario of Nanobiofertilizer Research and Applications

Researchers are becoming more interested in the use of nanoparticles in agriculture due to the quick results they provide. Nanoparticles can be combined with other compounds to make smart fertilizer, but little research has been reported. Recently, few articles have been published on the formulation and use of nanobiofertilizer, and it is a step toward the development of a new and more effective product for crops. Nanobiofertilizer has been prepared in Iran, and its application increased the amount of proline, carbohydrate, and anthocyanin in *Hibiscus sabdariffa* [124] and *Foeniculum vulgare* [144] under drought stress [145]. Jahangir et al. [125] carried out their experiments in Pakistan to show that the combined application of PGPR and Ag NPs increases the weight of *Allium cepa* facing high salinity. They also reported that the co-application increased the sugar content and the expression of some new proteins in plants. The chlorophyll, carotenoid, flavonoid, and phenolic content was also high in plants receiving the combined application of PGPR and Ag NPs. In India, the application of *Pseudomonas taiwanensis* and nano-gypsum improved bacterial diversity and their growth, enhanced the availability of nutrients, and increased soil moisture [146]. The application of this nanobiofertilizer also enhanced the activities of soil enzymes. Nano-gypsum supports the bacterial population and has positive effects on soil structure; it can be used as an alternative to agrochemicals. The collaborative action of *Pseudomonas taiwanensis* and nano-gypsum increases germination, root length, shoot length, and chlorophyll content in *Z. mays* [147]. Researchers of India have also shown that the use of nano-chitosan enhanced the colony number of PGPR. They act synergistically and improve the germination and growth attributes of *Z. mays* [148]. The use of nanocomposite biofertilizer on *T. aestivum* and *Cicer arietinum* enhanced plant biomass, crop yield, and plant resistance against biotic stress [149].

The improper and high dose application of Zn is toxic for plants and causes soil pollution. The collaborative work of Chinese and Pakistani researchers shows that nanobiofertilizer (having Zn nanoparticles) reduces the toxic effects of Zn and is an effective fertilizer with a slow-release property. It increases the number of leaves, plant height, 1000-grain weight, cob weight, and biological weight by 35%, 15%, 10%, 4%, and 42%, respectively. It also enhances the rate of photosynthesis, stomatal conductance, and transpiration [150]. The simultaneous application of *Mesorhizobium* sp., *Pseudomonas* and nano-silica improves the structure and fertility of the soil. They also act as a plant growth promoter and increase the fresh and dry weight of plants [151]. In Egypt, an experiment was performed to evaluate the effect of sole and combined application of Fe + Zn NPs and organic fertilizer on *Phaseolus vulgaris*. The results show that the combined application of Fe + Zn NPs and organic fertilizer had significant effects on plant growth and improved pods’ physical and nutritional quality. The analysis of soil at the end of the experiment showed that they also improved the quality of soil and made it more fertile [152]. The researchers of Iraq combined biofertilizer with the nano-element mixture and analyzed their effect on two varieties of *P. vulgaris*. The interaction of these biofertilizer-nano-scale elements significantly enhanced nutrient uptake, nodule formation, nitrogenase activity, the number of roots per plant, and amino acid content [153].

Nanobiofertilizer is being synthesized at various places in the world. Still, there is a need to study the effect of nanobiofertilizer on plants at the molecular level [154]. However, there is still a need to prepare a good nanobiofertilizer on a large scale, and nanobiofertilizer should be commercialized.

## 7. Constraints and Future Perspectives of Nanobiofertilizer

The use of nanobiofertilizer is expected to grow for the development of agriculture. However, the formulation should be environment-, plant-, and human-friendly. Although some articles report the formation of nanobiofertilizer, there is no detailed research about the mechanism of nanobiofertilizer for improving the growth of plants. The constituents of nanobiofertilizer include nanoparticles and biofertilizers. The use of nanoparticles is not yet common, and they are still on a trial basis [155,156]. To date, less research has been done on the development and application of nanobiofertilizer. There is a need to form nanobiofertilizer by properly merging nanoparticles and biofertilizer. Our recommendation for the future is as follows:(1)There is a need to use advanced equipment and protocols to formulate nanobiofertilizer with good quality, a longer shelf life, a low cost, and ease of use.(2)Detailed investigation of nanobiofertilizers should be carried out to evaluate their effects on human health.(3)A multifunctional nanobiofertilizer should be developed, which will be adequate for several crops.(4)An economic evaluation of nanobiofertilizer should be conducted. The nanobiofertilizer should be tested for its compatibility in various types of soil and environments.(5)Industries should collaborate to perform large-scale production, and field trials should be carried out.(6)There is a need to develop awareness among farmers about the side effects of chemical fertilizer and how nanobiofertilizer will reduce the cost and provide long-lasting effects.

## 8. Conclusions

The use of chemical fertilizer is disturbing the ecosystem and causing ill effects on human health. Nanobiofertilizers improve the growth of plants and increase their nutritional quality, productivity, shelf life, and resistance mechanisms against biotic and abiotic stress factors. The application of nanobiofertilizers maintains the level of nutrients in the soil and enhances the growth and yield of crops by activating various mechanisms. Nanobiofertilizers play a significant role in improving the quantity and quality of products. A very low concentration of nanobiofertilizer yeilds good results, and the chance of their bioaccumulation in the soil is very low, as they travel and work more efficiently. They enhance soil fertility and embody a unique combination with positive interactions as compared to a single dose of biofertilizer or nanoparticles. Our literature survey showed that researchers worldwide are working on nanoparticles and biofertilizers, though less work is being done on the formulation and application of nanobiofertilizer. Their combination provides better results as compared to their alone treatment. There is a lack of research indicating the development and mechanism of action of nanobiofertilizer at the molecular level in plants. The mutual interaction of nanoparticles and biofertilizer should be optimized in the form of nanobiofertilizer to control delivery and release at the target site. There is a need to develop a more productive nanobiofertilizer, and it should be applied at the practical level. Its effectiveness is related to the use of innovative approaches and detailed experimentations. We must gain an understanding of the mechanism used by nanobiofertilizer for agricultural development.

## Figures and Tables

**Figure 1 nanomaterials-12-00965-f001:**
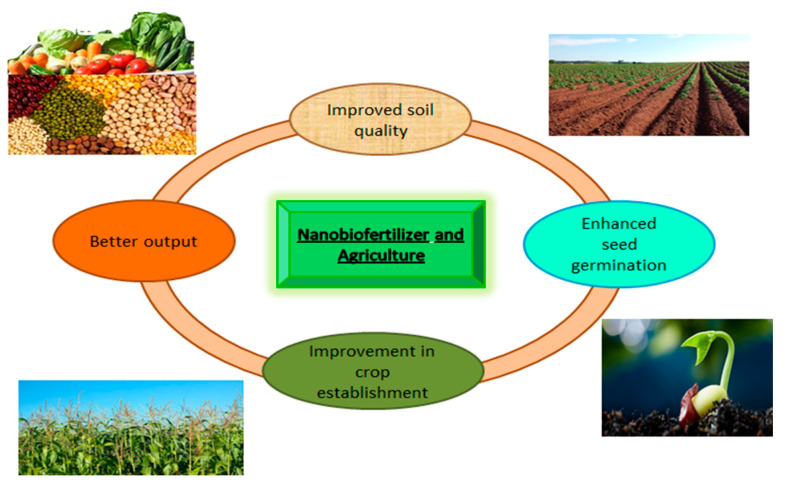
The basic role of nanobiofertilizer.

**Figure 2 nanomaterials-12-00965-f002:**
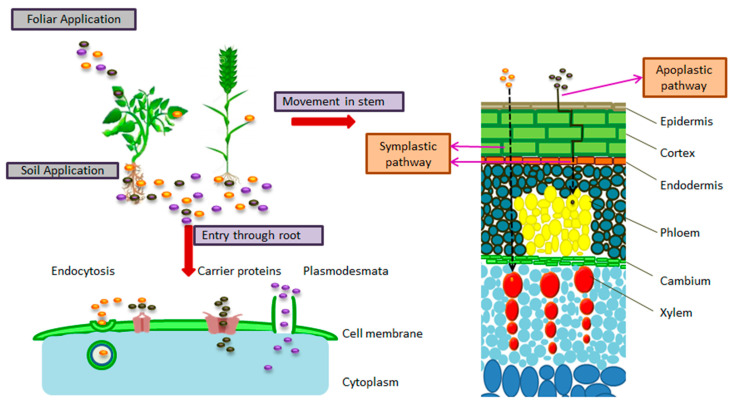
Uptake and movement of nanoparticles in plants.

**Figure 3 nanomaterials-12-00965-f003:**
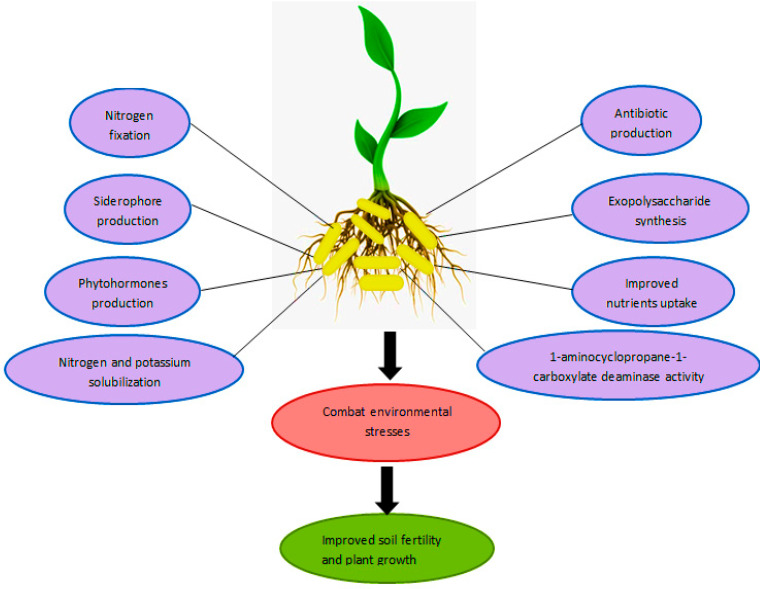
The role of plant-growth-promoting rhizobacteria in plant growth promotion and stress tolerance. Nitrogen fixation, mineral solubilization, and production of siderophore and phytohormones by PGPR promote plant growth. Production of antibiotics and other metabolites helps in the biocontrol of phytopathogens. Synthesis of exopolysaccharides, osmolytes, and antioxidants helps in stress tolerance.

**Figure 4 nanomaterials-12-00965-f004:**
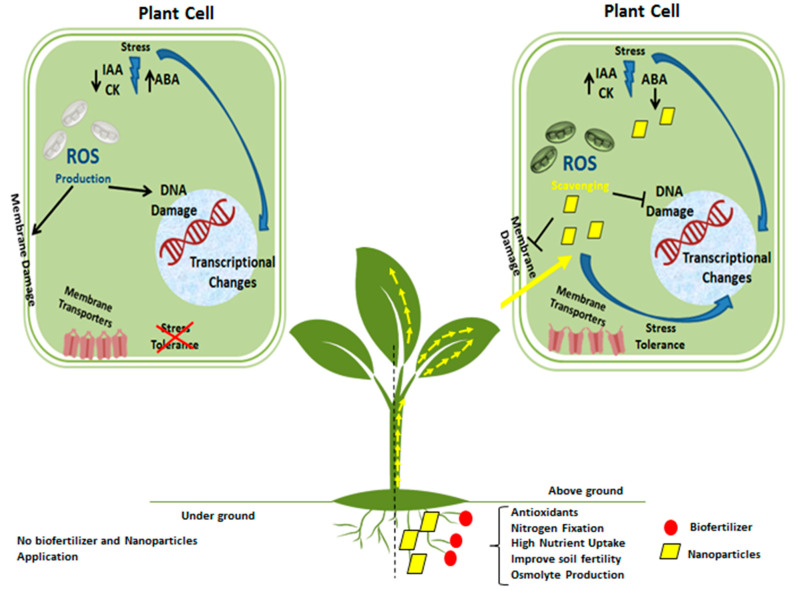
Role of nanobiofertilizer on the plant under stress. Under stress conditions, reactive oxygen species (ROS) are produced that damage the plant cell organs. The application of nanobiofertilizers aids in nitrogen fixation, nutrient solubilization, improved soil fertility, and the production of antioxidants and osmolytes. These antioxidants and osmolytes help the plant to scavenge ROSs.

**Table 1 nanomaterials-12-00965-t001:** Effect of nanoparticles on gene expression in plants.

Nanoparticles	Plants	Stress	Effect on Genes Expression	References
Silicon nanoparticles	*Solanum lycopersicum*	Salinity	Upregulation of NCED3, TAS14, CRK1, and AREB genes Downregulation of MAPK3, DDF2, MPAK2, ERF3, APX2, and RBOH1 genes	[87]
	*Oryza sativa*	Salinity	Upregulation of LSi1 and LSi2 genes	[88]
	*Oryza sativa*	Biotic stress	Upregulation of A2WZ30, B8AP99, A2YNH4, B8AZZ8, Fo2, B8B. B8A9F5, B8BF84. B8BHM9, and 2XRR2 genes	[89]
Copper nanoparticles	*Piper nigrum*	-	Upregulation of miR159 Downregulation of MVK gene	[90]
Iron nanoparticles	*Oryza sativa*	Cd and drought stress	Downregulation of OsLCT1, OsHMA3, and OsHMA2 genes	[91]
Silver nanoparticles	*Nigella sativa*	-	Upregulation of PAL and CHS genes	[92]
	*Arabidopsis thaliana*	-	Upregulation of 286 genes Downregulation of 81 genes	[93]
	*Arabidopsis thaliana*	-	Upregulation of PCS, GS, GR, and GSTU12 genes	[94]
	*Solanum lycopersicum*	-	Upregulation of PAL and EIX genes	[95]
	*Arabidopsis thaliana*	-	Upregulation of 438 genes	[96]

**Table 2 nanomaterials-12-00965-t002:** Role of nanobiofertilizer in different plants under control and stressed conditions.

Nanobiofertilizer	Plants	Conditions	Plants Responses	References
Nano-chelated B and Zn + Biofertilizer	*Zea mays*	Drought stress	✓ Improved morphological and physiological attributes including leaf area, ear diameter, ear length, chlorophyll content, and relative water content ✓ Enhanced yield attributes	[125]
Oxide NPs of Fe-Zn + Biofertilizer (*Azospirillum*, *Pseudomonas* and, *Azotobacter*)	*Triticum aestivum*	Stress and nonstress conditions	✓ Increased soluble sugar, proline, and enzymatic activities ✓ Enhanced the yield to 88% as compared to control (untreated drought exposed plants)	[126]
ZnNPs + Biofertilizer (*Rhizobium*) + Organic fertilizer	*Phaseolus vulgaris* L.	Normal	✓ Increased plant height, biomass leaf area, and number of leaves per plant ✓ Enhanced nutrient uptake, pod yield, and carbohydrate and protein content in pods	[127]
AgNPs + Biofertilizer (Nitroxin)	*Solanum tuberosum* L.	Normal	✓ Improved number of tubers per plant, diameter of tubers, and weight of tubers	[128]
TiNPs + Biofertilizer (*Azospirillum brasilense*, *A. caulinodans* and, *Azotobacter chroococcum*)	*Triticum secale*	Cadmium stress	✓ Increased chlorophyll content, relative water content, 1000-grain weight, and grain yield ✓ Decreased leaf Cd and seed Cd content	[129]
Nano-fertilizer + Biofertilizer (*Azetobacter*)	*Sorghum bicolor* (L.) Moench.	Normal	✓ Increased chlorophyll content, carotenoid content, and carbohydrate content	[130]
Zero-valen FeNPs + Biofertilizer (compost and biochar)	*Brassica juncea* L.	Heavy metal stress	✓ Improved soil cation exchange capacity, total nitrogen, total carbon, phosphorus, and pH ✓ Enhanced plant biomass and height	[131]
Acylated homoserine coated Fe-carbon nanofibres + Biofertilizer (*Panebacillus polymyxa*)	*Cicer aretianum* L. and *T. aestivum*	Biotic stress	✓ Improved biomass, length, and chlorophyll content ✓ Development of resistance against pathogens	[132]
Zero valent FeNPs + Biofertilizer (*Pseudomonas fluorescens*)	*Trifolium repens*	Antimony stress	✓ Improved plant growth and phytoremediation potential	[133]
Nanozeolite + Biofertilizer (*Bacillus* spp.)	*Z. mays*	Normal	✓ Enhanced plant length, chlorophyll, and protein content	[134]
ZnO NPs + biofertilizer (composted biochar farmyard manure)	*T. aestivum*	Cd stress	✓ Enhanced biomass, photosynthetic pigments, antioxidants, and yield of the plant	[135]

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
