# Peer review of "Improvement of Plant Responses by Nanobiofertilizer: A Step towards Sustainable Agriculture"

_nanomaterials, 2022, doi:10.3390/nano12060965_

Round 1
Reviewer 1 Report
The authors have made a tiny effort to add at least two more bridging paragraphs, which ... help. The English is clearly non-native and they did not want to change it or cannot, to a more native style. But, it is understandable enough, just doesnt read as smoothly as it could to my ears. I was hoping the rejection would spur them on to have a closer look at it.
In my personal opinion more negative aspects (counter arguments) could still be added, as the perspective that these particles only do good seems too simplistic an explanation. But, the current momentum is a positive note, so be it. This is the authors' work, not mine.
The first added paragraph was written hastily, has run-on sentences and its logic is not finished. Please view the png.

Author Response
Reviewers 1 Report
The authors have made a tiny effort to add at least two more bridging paragraphs, which ... help. The English is clearly non-native and they did not want to change it or cannot, to a more native style. But, it is understandable enough, just doesn’t read as smoothly as it could to my ears. I was hoping the rejection would spur them on to have a closer look at it.
In my personal opinion more negative aspects (counter arguments) could still be added, as the perspective that these particles only do good seems too simplistic an explanation. But, the current momentum is a positive note, so be it. This is the authors' work, not mine.
Authors’ Response: The authors are thankful to the reviewer for such a critical reviewing that helped in the significant improvement of the manuscript. This topic is relatively new and less information is available. We have tried to go through all the relevant literature again and tried to add as much information as available. We have also tried to improve language at our best. As suggested, the manuscript has been revised thoroughly to improve the flow, English language and grammar. We hope that you consider this all and accept the manuscript.
The first added paragraph was written hastily, has run-on sentences and its logic is not finished. Please view the png.
Authors’ Response: As suggested, corrections have been made in the last para of subheading 2.1. Nano-particles

Reviewer 2 Report
The work presented by Akhtar et al. and titled "Manipulation of Plant Responses by Nanobiofertilizer in the Changing Climate" was pleasant to read. The work is well-written and easy to follow and, in my opinion, ready for publication. However, prior to that, I would like the author to reflect on the title of the paper and how it anticipates that the work will focus on how the implementation of nanoparticles in agriculture can have an effect on climate change mitigation (It is correct?). However, reading the article, I found that the main topic covered relates to the environmental aspects of the nanoparticles use, i.e. how they can reduce the load of substances and trace elements in the soil and agricultural environment, thus reducing health risks. Hence, I would like the author to better explain the choice of title and (perhaps) propose a different, more precise one.
In addition, it is well-known as also wood distillate is considered a promising green bio-based product to enhance plant productivity of horticultural interest. But, does the author think that such a product may have a position in section 2.2?
Reviewer 2 Report
The work presented by Akhtar et al. and titled "Manipulation of Plant Responses by Nanobiofertilizer in the Changing Climate" was pleasant to read. The work is well-written and easy to follow and, in my opinion, ready for publication. However, prior to that, I would like the author to reflect on the title of the paper and how it anticipates that the work will focus on how the implementation of nanoparticles in agriculture can have an effect on climate change mitigation (It is correct?). However, reading the article, I found that the main topic covered relates to the environmental aspects of the nanoparticles use, i.e. how they can reduce the load of substances and trace elements in the soil and agricultural environment, thus reducing health risks. Hence, I would like the author to better explain the choice of title and (perhaps) propose a different, more precise one.
Authors’ Response: Agreed to the suggestion and the title has been changed as per the suggestion.
In addition, it is well-known as also wood distillate is considered a promising green bio-based product to enhance plant productivity of horticultural interest. But, does the author think that such a product may have a position in section 2.2?
Authors’ Response: We have only added the data that was relevant to nanoparticles and PGPR, which were mostly used as biofertilizer and the research which has been done on their combined application so we did not focus on other aspects which did not relate much.

Reviewer 3 Report
The manuscript entitled “Manipulation of Plant Responses by Nanobiofertilizer in the Changing Climate” This work is merit for publication at Nanomaterials after some major modification. So I have some points that may help to improve the work as follows:
1-Abstract is good but need more explain about the main aim of work
2- The introduction should be extended to discuss the hypothesis and research questions in details. Additionally, the introduction should cover the recent literature related to this subject.
3- The conclusion
A section for conclusions need more explain.
Line 126-160, please rephrase it.
Line 226- 263, please rephrase it.
Line 362-388, please rephrase it.
Line 468-513, please rephrase it.
4- English writing should be checked by a native English speaking expert.
Reviewer 3 Report
The manuscript entitled “Manipulation of Plant Responses by Nanobiofertilizer in the Changing Climate” This work is merit for publication at Nanomaterials after some major modification. So I have some points that may help to improve the work as follows:
1-Abstract is good but need more explain about the main aim of work
Authors’ Response: The purpose of writing this review articles has been explained in last para of abstract (Pg 1).
2- The introduction should be extended to discuss the hypothesis and research questions in details. Additionally, the introduction should cover the recent literature related to this subject.
Authors’ Response: The hypothesis and research questions have been explained and some recent data has been added in the second last para of introduction (Page 2, 3).
3- The conclusion
A section for conclusions need more explain.
Authors’ Response: We have tried to make the abstract catchy. More text has been added
Line 126-160, please rephrase it.
Authors’ Response: [2.1. 1. Silicon Nanoparticles (SiNPs) page 4, 5], It has been rephrased.
Line 226- 263, please rephrase it.
Authors’ Response: [2.1.3. Copper Nano-particles (CuNPs) page 6, 7], it has been rephrased.
Line 362-388, please rephrase it.
Authors’ Response: [3. Formulation of Nanobiofertilizer, page 10,11], it has been rephrased.
Line 468-513, please rephrase it.
Authors’ Response: [6. Current Scenario of Nanobiofertilizer, page 13, 14] it has been rephrased.
4- English writing should be checked by a native English speaking expert.
Authors’ Response: Whole article have been checked for English through Licensed copy of Grammarly software.

Reviewer 4 Report
Akhtar et al. deal with the potential of nano-biofertilizers in improving plant growth and productivity under non-optimal conditions. Nanomaterials and bio-fertilizers are two emerging fields, but their combined use indeed has received limited attention. It is an interesting manuscript. The authors are requested to deal with the following comments within the manuscript (rather than the response letter):
(1) The concepts of nano-biofertilizers/ green synthesis need to be better described in the Abstract. The promising potential in the postharvest sector also ought to be shortly discussed.
(2) Line 35: plant species in place of plants
(3) Lines 42/ 44 & constraints (section 7): please define the terms quality/ nutritional quality. Quality is a very broad terms, which involved several features, the importance of which depends on the intended use (Paschalidis et al., 2021 Sustainability 13, 14030). A very common quality criterion is the content of secondary metabolites, which is known to be increased by stress. The metabolites exhibit strong human heath promoting features. These metabolites are also increased by nano-biofertilizers. In this regards, nano-biofertilizers promote human health by stimulating the content of health-promoting compounds.
(4) Title of the paper refers to climate change. However, little text is devoted on climate change in the manuscript. For instance, UV levels are projected to rise globally owing to anthropogenic climate change, while UVA radiation exerts a large impact on agricultural ecosystems (Mumivand et al., 2022 Horticulturae 8, 31). In natural habitats, elevated UV levels mostly coincide with high temperature and drought events (Mumivand et al., 2022 Horticulturae 8, 31). In addition, the use of agricultural land is often limited by excess or shortage of a range of nutrients in arid parts of the world, as well as in areas hosting heavy industrial activity or employing lower quality irrigation water (Chatzistathis et al., 2021 Agronomy 11, 759). In arid areas, these limitations are expected to be intensified by climate change through increased frequency of drought events.
(5) Line 79: please take care when mentioning organic certification label. Is this what you mean? If not, please remove it!
(6) Lines 111 & elsewhere: the factors affecting efficiency of nano-particles (size, shape, etc.) are repeated at several places throughout the manuscript. You need to mention them once. Please stress that the size of nano-particles is often inversely related to their uptake.
(7) You need to mention that foliar application requires much smaller volume than soil application. This sizeable difference sets spraying not only more inexpensive for the agricultural industry, but also minimizes the environmental impact (Ahmadi-Majd et al., 2021 Journal of Horticultural Sciences and Biotechnology DOI: 10.1080/14620316.2021.1993755)
(8) constraints (section 7): Nanomaterials show high potential as a viable means of improving plant growth and productivity, as thoroughly discussed in this manuscript. Note that in very recent work, their potential for employment in the postharvest sector is explored with very promising results too (Costa et al. 2020 Frontiers in Plant Science, 11, 584698; Ahmadi-Majd et al., 2021 Journal of Horticultural Sciences and Biotechnology DOI: 10.1080/14620316.2021.1993755; Ahmadi-Majd et al., 2022 Chemical and Biological Technologies in Agriculture 9, 15).
(9) The term green synthesis is repeated in several places throughout the manuscript. The authors need to clearly define what is meant by this term.
(10) Some positive effects of silver nano-particles are presented. However, authors should not neglect mentioning that Ag is a heavy metal, and its use is strongly prohibited in most countries. Therefore, it can be mostly used/ introduced as a laboratory example.
(11) The source of bio-fertilizer is repeated throughout the manuscript, where different materials are often reported. You need to group them, and mention all of them once.
(12) A grouping of the general effects is desirable, like authors attempt to do in Table 2. Non-enzymatic (mention examples) and enzymatic (mention examples) antioxidant defense elements are activated. Water relations are improved through improved water uptake, since water loss mostly increased. The increased water uptake was the result of enhanced aquaporin function.
(13) Authors neglect to mention what is the appropriate stage of application. Optimally, does the application involve a single time? Does it work in fully-grown plants? Does it work during the postharvest phase? Does it work in seeds?
(14) In edible crops, adverse effects of nanomaterials on human health and environment have been suggested. In this regard, no laxity in application ought to be tolerated, and disposal issues ought to be deliberated before commercial use (Ahmadi-Majd et al., 2022 Chemical and Biological Technologies in Agriculture 9, 15).
Reviewer 4 Report
Akhtar et al. deal with the potential of nano-biofertilizers in improving plant growth and productivity under non-optimal conditions. Nanomaterials and bio-fertilizers are two emerging fields, but their combined use indeed has received limited attention. It is an interesting manuscript. The authors are requested to deal with the following comments within the manuscript (rather than the response letter):
- The concepts of nano-biofertilizers/ green synthesis need to be better described in the Abstract. The promising potential in the postharvest sector also ought to be shortly discussed.
Authors’ Response: Abstract has been modified.
(2) Line 35: plant species in place of plants
Authors’ Response: Corrected as per the suggestion.
- Lines 42/ 44 & constraints (section 7): please define the terms quality/ nutritional quality. Quality is a very broad terms, which involved several features, the importance of which depends on the intended use (Paschalidis et al., 2021 Sustainability 13, 14030). A very common quality criterion is the content of secondary metabolites, which is known to be increased by stress. The metabolites exhibit strong human heath promoting features. These metabolites are also increased by nano-biofertilizers. In this regards, nano-biofertilizers promote human health by stimulating the content of health-promoting compounds.
Authors’ Response: The data related to quality have been added in section “5. Nanobiofertilizer and Plants Responses”
- Title of the paper refers to climate change. However, little text is devoted on climate change in the manuscript. For instance, UV levels are projected to rise globally owing to anthropogenic climate change, while UVA radiation exerts a large impact on agricultural ecosystems (Mumivand et al., 2022 Horticulturae 8, 31). In natural habitats, elevated UV levels mostly coincide with high temperature and drought events (Mumivand et al., 2022 Horticulturae 8, 31). In addition, the use of agricultural land is often limited by excess or shortage of a range of nutrients in arid parts of the world, as well as in areas hosting heavy industrial activity or employing lower quality irrigation water (Chatzistathis et al., 2021 Agronomy 11, 759). In arid areas, these limitations are expected to be intensified by climate change through increased frequency of drought events.
Authors’ Response: Data related to climate change have been added on page 2 and the above mentioned literature has been cited.
- Line 79: please take care when mentioning organic certification label. Is this what you mean? If not, please remove it!
Authors’ Response: Line 79 (page 2) have been removed.
(6) Lines 111 & elsewhere: the factors affecting efficiency of nano-particles (size, shape, etc.) are repeated at several places throughout the manuscript. You need to mention them once. Please stress that the size of nano-particles is often inversely related to their uptake.
Authors’ Response: The repetition of “size “have been removed from the article.
(7) You need to mention that foliar application requires much smaller volume than soil application. This sizeable difference sets spraying not only more inexpensive for the agricultural industry, but also minimizes the environmental impact (Ahmadi-Majd et al., 2021 Journal of Horticultural Sciences and Biotechnology DOI: 10.1080/14620316.2021.1993755)
Authors’ Response: page 9; Data related to importance of foliar spray has been added.
(8) constraints (section 7): Nanomaterials show high potential as a viable means of improving plant growth and productivity, as thoroughly discussed in this manuscript. Note that in very recent work, their potential for employment in the postharvest sector is explored with very promising results too (Costa et al. 2020 Frontiers in Plant Science, 11, 584698; Ahmadi-Majd et al., 2021 Journal of Horticultural Sciences and Biotechnology DOI: 10.1080/14620316.2021.1993755; Ahmadi-Majd et al., 2022 Chemical and Biological Technologies in Agriculture 9, 15).
Authors’ Response: We have cited the relevant reference ‘Ahmadi-Majd et al., 2021 Journal of Horticultural Sciences and Biotechnology’ at page 9.
We only mentioned about the nanoparticles that were synthesized by using plants and that were used in combination with biofertilizer
(9) The term green synthesis is repeated in several places throughout the manuscript. The authors need to clearly define what is meant by this term.
Authors’ Response: [2.1,1. Silicon nanoparticles SiNPs; page 5] the term green synthesis has been explained.
(10) Some positive effects of silver nano-particles are presented. However, authors should not neglect mentioning that Ag is a heavy metal, and its use is strongly prohibited in most countries. Therefore, it can be mostly used/ introduced as a laboratory example.
Authors’ Response: Yes, Silver is a heavy metal and we have mentioned that nanobiofertilizer should be evaluated first for their effects on human health, on soil and environment. It will reduce the chances of toxicity. (in section; Constrains and future perspective of nanobiofertilizer)
(11) The source of bio-fertilizer is repeated throughout the manuscript, where different materials are often reported. You need to group them, and mention all of them once.
Authors’ Response: Corrections have been made. The details about biofertilizer are mentioned in section 2.2.
(12) A grouping of the general effects is desirable, like authors attempt to do in Table 2. Non-enzymatic (mention examples) and enzymatic (mention examples) antioxidant defense elements are activated. Water relations are improved through improved water uptake, since water loss mostly increased. The increased water uptake was the result of enhanced aquaporin function.
Authors’ Response: The correction has been made on page 12. Examples of antioxidants have been given.
(13) Authors neglect to mention what is the appropriate stage of application. Optimally, does the application involve a single time? Does it work in fully-grown plants? Does it work during the postharvest phase? Does it work in seeds?
Authors’ Response: we did not find literature where the effect of nanobiofertilizer have been explored at different stages so how we can conclude which stage is best. It has been used on different plants and gave good results but still there is a need to define the stage of application.
(14) In edible crops, adverse effects of nanomaterials on human health and environment have been suggested. In this regard, no laxity in application ought to be tolerated, and disposal issues ought to be deliberated before commercial use (Ahmadi-Majd et al., 2022 Chemical and Biological Technologies in Agriculture 9, 15).
Authors’ Response: It has been mentioned point 2 under the heading 7. Constraints and Future Perspectives of Nanobiofertilizer.

Round 2
Reviewer 3 Report
The authors have made changes to the manuscript, so I consider it can be accepted for publication.
Reviewer 4 Report
authors did excellent work in dealing with my comments
the manuscript is now appropriate for publication
This manuscript is a resubmission of an earlier submission. The following is a list of the peer review reports and author responses from that submission.
Round 1
Reviewer 1 Report
There are many reviews out there with this topic. It was difficult to see where this review differs, although I guess the authors want to emphasize nanobiofertilizers (as the title suggests). If the review is to be kept, it needs to be improved by taking a clearer approach to their arguments, organize the review in three steps: nanoparticle fertilizers, biofertilizers (only 1 paragraph) and then the combination nano-bio-fertilizers.
Secondly, just listing what other reviews or papers said is important is not good interpretation. Either quote the other reviews/studies headfirst, or, synthesize your own ideas. Make this clear to the reader which is which.
Thirdly, while the authors did the right thing in writing it on their own and not recycling other sentences out there, the English needs to be more fluid. I gave some examples, but an idea is to reduce the number of words. This will force you to be more concise. Be careful with adjective/adverbial placement, and place main actions/ideas up front. Reads better, is more clear. I am sure they can do it. It might take two to three more rounds, but it is possible. But first, focus the arguments and support them with data (studies). Only then can we scientifically criticize and appreciate their work.
I added some, but only some, help in English and story; the authors still need to go over it entirely.

Author Response
Response to Comments of First Reviewer
Comments and Suggestions for Authors
There are many reviews out there with this topic. It was difficult to see where this review differs, although I guess the authors want to emphasize nanobiofertilizers (as the title suggests). If the review is to be kept, it needs to be improved by taking a clearer approach to their arguments, organize the review in three steps: nanoparticle fertilizers, biofertilizers (only 1 paragraph) and then the combination nano-bio-fertilizers.
Author’s response: The authors are thankful for the valuable suggestion and comments of the reviewer which helped in the improvement of the manuscript. The whole manuscript has been revised keenly in the light of suggestions proposed. The review has been organized as per the recommended layout i.e. nanoparticles, 1 paragraph of biofertilizer, and nanobiofertilizer; Page 5.
Secondly, just listing what other reviews or papers said is important is not good interpretation. Either quote the other reviews/studies headfirst, or, synthesize your own ideas. Make this clear to the reader which is which.
Author’s response: Your valuable suggestions has been well taken to improve the MS. We have tried to make a link and create flow in the text.
Thirdly, while the authors did the right thing in writing it on their own and not recycling other sentences out there, the English needs to be more fluid. I gave some examples, but an idea is to reduce the number of words. This will force you to be more concise. Be careful with adjective/adverbial placement, and place main actions/ideas up front. Reads better, is more clear. I am sure they can do it. It might take two to three more rounds, but it is possible. But first, focus the arguments and support them with data (studies). Only then can we scientifically criticize and appreciate their work.
I added some, but only some, help in English and story; the authors still need to go over it entirely.
Author’s response: Thanks for your valuable suggestion. We have revised the manuscript and tried to remove the mistakes. We have tried to make it concise. We have carefully read the whole manuscript and have tried to remove all mistakes.
PDF file comments/response
WARNING: English corrections are only sporactically done. This does not mean the rest is fine. The English needs to be re-worked at least 2-3x to get the flow and meaning across better. It is very bumpy right now. I am a native speaker.
Author’s response: We have tried to carefully read the whole manuscript and remove the mistakes.
The Abstact assumes that the reader is familar with the topic. Since this is a "new topic", term definitions need to be stated clearly before proceeding.
Author’s response: Improvements have been made in the abstract, all relevant terms have been defined; page 1.
something different here. (plant cell under stress?)... and your round , cell wall free plant cell is a bit odd... the cell wall and apoplastic space is also a large component of the environmental interaction
if no figure legend is going to be used, then the abbreviations need to be spelled out.
Author’s response: The cell shape has been corrected and we have added cell wall and membrane to plant cell. The abbreviations have been written in full form in the figure; page 3
Introduction; very wordy, ... try to get your sentences to be more fluid by requiring less pauses between conceptual linkages
Author’s response: We have tried to improve introduction. The highlighted sentence has been removed; page 3.
Parent material???
Author’s response: page 3; Sometimes, parent rock material releases a large amount of salts through their weathering and releases salts which cause salinity and heavy metals accumlation.
ions and atoms are still ions and atoms. Smallness is not really a good argument.
Author’s response: page 4; The word ‘smallness’ have been removed.
huh? define - explain.
Author’s response: page 4; The term encapsulation has been defined.
HOW? I had to look at other reviews to substantiate your claim. You dont have to write another paragraph, but you need to give the reader and idea of how (one or two proposed mechanisms) this decrease is supposed to happen. less belief -> more argument.
Author’s response: page 4; The application of nanobiofertilizer is highly preferred upon chemical fertilizer and it has also been documented in the following articles. The use of nanobiofertilizer overcome the demands of chemical fertilizers and this has been documented in the section ‘Nanobiofertilizer application and plant responses.
(Kumari and Singh, 2020)(Fallah Nosrat Abad et al., 2020)
Fallah Nosrat Abad, A., Aftab Talab, A., and Shariati, S. (2020). Increasing the yield of maize and improving the chemical and biological properties of saline calcareous soil using a combination of nano-biofertilizer and cattle manure. Cereal Res. 10, 259–271. doi:10.22124/CR.2021.17715.1625.
Kumari, R., and Singh, D. P. (2020). Nano-biofertilizer: An Emerging Eco-friendly Approach for Sustainable Agriculture. Proc. Natl. Acad. Sci. India Sect. B - Biol. Sci. 90, 733–741. doi:10.1007/S40011-019-01133-6.
- Constituents of Nanobiofertilizer; right now "it" refers to application not biofertilizer AND this definition should have come a lot earlier.
Author’s response: page 11; The sentence has been rephrased.
all types of bio-fertilizer produce this? Siderophore
Author’s response: page 11; No. This sentence has been restructured.
example ...! of Phytohormones
Author’s response: page 11; Addition has been done in the sentence ‘They also synthesize phytohormones like indole acetic acid and cytokinin that act synergistically….
very bold statement; can it be substantiated?
Author’s response: Page 11; The sentence has been rephrased.
back up with literature or facts or both
Author’s response: page 11; A lot of literature has been cited and details have been mentioned in the subheadings of this section.
new section (paragraph) – nanoparticles don't mix them. place definitions and opening clarifications at the beginning of a paragraph (i.e. last sentence here, after the first sentence).
Author’s response: Page 11; The definition has been added and arrangement has been changed accordingly.
Figure; better spelled out fully with abbreviation
why different colors? are the lines pointing to the soil or the bacteria or both?
Author’s response: page 12; The abbreviations have been written in full form. The bacteria color has been changed and we remove the soil to make the figure clear as the lines are pointing toward bacterial strains.
https://www.nature.com/articles/s41598-020-77059-1
https://www.ncbi.nlm.nih.gov/pmc/articles/PMC6027514/
to bolster your arguments
Author’s response: page 6; The articles have been cited to support the statements.
2.1. Silicon Nanoparticles (SiNPs); This paragraph is just a bullet list. While it is not pleasant to read off a bunch of properties, it is also unclear if all the statements are actually encapsulated in the references provided. Flip it around, refer to the studies, refer to the evidence.
Author’s response: page 7; All the references in the mentioned paragraph have been rechecked and we have mentioned the data of these articles. The paragraph has been restructured to create flow in the data.

Reviewer 2 Report
The submitted manuscript reviewed the state of the art on nanobiofertilizers, by restricting the attention to the combination nanoparticle-PGRR. In order to properly cover the field, I would suggest to make the following modifications:
- please add information on the generic term for nanobiofertilizer, that actually does not include only the cited combination, but even different phytoactive agents and different nanoparticles; as an example, biofertilizer is even biofertilizer derived from biomass;
- additionally, it is requested to add information on the use of PGRR when combined with other nanoparticles other than metallic or metal oxide ones;
- it is required to better explain the graphical abstract that the authors introduced in the text, that was not neither cited in the main text.
Author Response
Response to comments of Reviewer 2
Comments and Suggestions for Authors
The submitted manuscript reviewed the state of the art on nanobiofertilizers, by restricting the attention to the combination nanoparticle-PGRR. In order to properly cover the field, I would suggest to make the following modifications:
Author’s response: We are thankful for your worthy comments. We have tried to make changes accordingly.
please add information on the generic term for nanobiofertilizer, that actually does not include only the cited combination, but even different phytoactive agents and different nanoparticles; as an example, biofertilizer is even biofertilizer derived from biomass;
Author’s response: The main property and definition of nanobiofertilizer has been added in the start of section ‘Formulation of biofertilizer’. Synthesis of nanobiofertilizer by using other organic material (biofertilizer) have been given at the end of this section.
There are many articles which use the term nanobiofertilizer but they only prepare nanoparticles (reduce size of salts to nm) by using plant extract, microbes and algae and call it nanobiofertilizer but it’s something else. Their end products were only nanoparticles, which were applied on crops. So I selected only those articles which use a combination of Biofertilizer and nanoparticles and which fall in the proper definition of nanobiofertilizer i.e. it’s a hybrid of nanoparticles and biofertilizer.
The current definition of biofertilizer has been added in the section 2.2.
additionally, it is requested to add information on the use of PGRR when combined with other nanoparticles other than metallic or metal oxide ones;
Author’s response: Less literature is available on the application of nanobiofertilizer but ewe have added some relevant material other than metallic oxides in Table 2.
- it is required to better explain the graphical abstract that the authors introduced in the text, that was not neither cited in the main text.
Author’s response: page 4; Graphical abstract has been cited and explained in the last para of introduction.
Response to comments of Reviewer 2
Comments and Suggestions for Authors
The submitted manuscript reviewed the state of the art on nanobiofertilizers, by restricting the attention to the combination nanoparticle-PGRR. In order to properly cover the field, I would suggest to make the following modifications:
Author’s response: We are thankful for your worthy comments. We have tried to make changes accordingly.
please add information on the generic term for nanobiofertilizer, that actually does not include only the cited combination, but even different phytoactive agents and different nanoparticles; as an example, biofertilizer is even biofertilizer derived from biomass;
Author’s response: The main property and definition of nanobiofertilizer has been added in the start of section ‘Formulation of biofertilizer’. Synthesis of nanobiofertilizer by using other organic material (biofertilizer) have been given at the end of this section.
There are many articles which use the term nanobiofertilizer but they only prepare nanoparticles (reduce size of salts to nm) by using plant extract, microbes and algae and call it nanobiofertilizer but it’s something else. Their end products were only nanoparticles, which were applied on crops. So I selected only those articles which use a combination of Biofertilizer and nanoparticles and which fall in the proper definition of nanobiofertilizer i.e. it’s a hybrid of nanoparticles and biofertilizer.
The current definition of biofertilizer has been added in the section 2.2.
additionally, it is requested to add information on the use of PGRR when combined with other nanoparticles other than metallic or metal oxide ones;
Author’s response: Less literature is available on the application of nanobiofertilizer but ewe have added some relevant material other than metallic oxides in Table 2.
- it is required to better explain the graphical abstract that the authors introduced in the text, that was not neither cited in the main text.
Author’s response: page 4; Graphical abstract has been cited and explained in the last para of introduction.

Round 2
Reviewer 1 Report
1. The abstract is written redundantly. Likely copy-paste error, but ...
2.The English is very bad. It needs to be revised properly. It is not worth accepting until the English is brought to a version that is acceptable.
This includes typos, misused expressions, grammar, etc. Way too many.
3. Figures should have proper legends, with title and a substantial description.
Author Response
REVIEWER # 1
AUTHORS RESPONSE: The authors are thankful for the valuable suggestion and comments which helped in the improvement of the manuscript. The whole manuscript has been revised keenly in light of suggestions proposed by reviewers. Your valuable suggestions are well taken to improve the MS. The pointwise response to each of your comment/suggestions is given below.
- The abstract is written redundantly. Likely copy-paste error, but ...
AUTHORS RESPONSE: We have tried to remove the mistakes from abstract.
2.The English is very bad. It needs to be revised properly. It is not worth accepting until the English is brought to a version that is acceptable.
This includes typos, misused expressions, grammar, etc. Way too many.
AUTHORS RESPONSE: The whole manuscript has been thoroughly checked for all language issues and grammatical mistakes. The manuscript has also been checked through professional language software of Grammarly.
- Figures should have proper legends, with title and a substantial description.
AUTHORS RESPONSE: Figure legends and descriptions have been added.

Reviewer 2 Report
The authors considered all the sugegsted modifications, so now the paper can be accepted for publication in its present form
Author Response
REVIEWER # 2
The authors considered all the suggested modifications, so now the paper can be accepted for publication in its present form
AUTHORS RESPONSE: We are thankful to the reviewer.
